# Navigation task and action space drive the emergence of egocentric and allocentric spatial representations

**Sandhiya Vijayabaskaran** *, **Sen Cheng**

Faculty of Computer Science, Ruhr University Bochum, Bochum, Germany

* sandhiya.vijayabaskaran@rub.de

**Data Availability Statement:** All source code used to produce the results and analyses presented in this manuscript is available at:

## Abstract

In general, strategies for spatial navigation could employ one of two spatial reference frames: egocentric or allocentric. Notwithstanding intuitive explanations, it remains unclear however under what circumstances one strategy is chosen over another, and how neural representations should be related to the chosen strategy. Here, we first use a deep reinforcement learning model to investigate whether a particular type of navigation strategy arises spontaneously during spatial learning without imposing a bias onto the model. We then examine the spatial representations that emerge in the network to support navigation. To this end, we study two tasks that are ethologically valid for mammals—guidance, where the agent has to navigate to a goal location fixed in allocentric space, and aiming, where the agent navigates to a visible cue. We find that when both navigation strategies are available to the agent, the solutions it develops for guidance and aiming are heavily biased towards the allocentric or the egocentric strategy, respectively, as one might expect. Nevertheless, the agent can learn both tasks using either type of strategy. Furthermore, we find that place-cell-like allocentric representations emerge preferentially in guidance when using an allocentric strategy, whereas egocentric vector representations emerge when using an egocentric strategy in aiming. We thus find that alongside the type of navigational strategy, the nature of the task plays a pivotal role in the type of spatial representations that emerge.

## Author summary

Most species rely on navigation in space to find water, food, and mates, as well as to return home. When navigating, humans and animals can use one of two reference frames: one based on stable landmarks in the external environment, such as moving due north and then east, or one centered on oneself, such as moving forward and turning left. However, it remains unclear how these reference frames are chosen and interact in navigation tasks, as well as how they are supported by representations in the brain. We therefore modeled two navigation tasks that would each benefit from using one of these reference frames, and trained an artificial agent to learn to solve them through trial and error. Our results show that when given the choice, the agent leveraged the appropriate reference frame to

https://github.com/sencheng/Emergent-Spatial-Representations-in-RL.

**Funding:** This work was supported by the Deutsche Forschungsgemeinschaft (DFG, German Research Foundation: https://www.dfg.de/) – project number 316803389 – through SFB 1280, project A14 The funders had no role in study design, data collection and analysis, decision to publish, or preparation of the manuscript.

**Competing interests:** The authors have declared that no competing interests exist.

solve the task, but surprisingly could also use the other reference frame when constrained to do so. We also show that the representations that emerge to enable the agent to solve the tasks exist on a spectrum, and are more complex than commonly thought. These representations reflect both the task and reference frame being used, and provide useful insights for the design of experimental tasks to study the use of navigational strategies.

## Introduction

Spatial navigation and its underlying neural mechanisms have been the subject of intense research for several decades. Broadly speaking, spatial navigation can be achieved using strategies that rely on one of two reference frames. An egocentric navigation strategy chooses actions in a reference frame centered on the agent, e.g. turn left, then go forward. By contrast, in an allocentric strategy, actions are referenced in the external space based on stable landmarks in the environment, e.g. go north, then turn west [1, 2]. Much research has been dedicated to the dissociation of the two strategies, since they are impacted differently by aging and disease [3–6], and due to interest in uncovering parallel memory systems that support spatial navigation [7]. In general, there are two ways to design tasks to behaviorally differentiate strategy use. Some tasks, such as the classic T-maze can be solved by animals using either an egocentric or an allocentric strategy. The presence of distal and intramaze cues enables testing for preferred strategy by manipulating these cues [8]. On the other hand, other tasks are expressly designed to elicit the use of a specific strategy. Among the most well-known of these tasks is a standard behavioral test of spatial memory—the Morris Water Maze and its dry variant, the cheese board maze [9, 10]. Although this task is intended to evoke an allocentric strategy, there is evidence that some animals can solve it using an egocentric view matching strategy, especially in the early stages of learning [11].

While the studies mentioned above are centered on the behavioral dissociation of the two strategies, other studies focus on identifying the brain regions that support the use of these strategies. Lesion studies in animals are frequently used to isolate brain areas responsible for the use of either strategy. One of the most well-known findings is that impairing the rodent hippocampus results in a deficit in the use of allocentric strategies in the Morris Water Maze [12–14]. However, the neural substrates of egocentric navigation strategies have received less attention, with the dorsal striatum [14, 15] and thalamus [16] being implicated. Interestingly, the striatum has also been implicated in the use of allocentric strategies, and the hippocampus in the use of egocentric strategies [14]. This shows that while these behavioral and lesion studies attempt to separate the use of the two strategies, in practice, they overlap, and real world navigation is likely a combination of the two, switching between them as required. This dynamic interplay between navigation strategies massively complicates their study.

A key question in neuroscience is how the use of allo- and egocentric strategies are related to spatial representations in the brain. At first glance, it seems intuitive to postulate that ego- and allocentric strategies are supported by ego- and allocentric neural representations, respectively. For instance, the allocentric coding of spatial locations by place cells [17] might drive allocentric navigation strategies and egocentric bearing-by-distance tuning to a visibly marked goal [18] might drive egocentric navigation strategies. However, there are three major issues with this view.

Firstly, both types of spatial representations are found in the hippocampus, which according to numerous studies, including some cited above, is thought to be the locus of the allocentric cognitive map. Similarly, spatial representations in the surrounding areas of the

hippocampus are mixed and include, for instance, allocentric grid cells [19] and border cells [20] in medial entorhinal cortex as well as egocentric representations of boundaries and objects in lateral entorhinal cortex [21]. These results suggest that the mapping of allo- and egocentric strategies to allo- and egocentric neural representations might be less tight than commonly postulated. Secondly, many neural recordings, such as those from place cells in the hippocampus, come from experiments where animals are not actually engaged in a concrete navigation task. Rather, they come from sessions where the animals are engaged in random foraging, or from runs on linear tracks where they are over-trained. As a result, it is difficult, if not impossible, to link the neural representations recorded from these experiments to the use of concrete navigation strategies. The importance of the task is underscored by evidence that spatial representations are modulated by elements of the task. In the Morris Water Maze, for instance, place cells tend to be clustered around the goal location [22, 23]. Similarly, goal location modulates spatial representation in bats, where cells in CA1 encode egocentric angles and distance to the goal [24]. Other studies have shown that the hippocampus represents task-relevant variables also in nonspatial domains [25, 26].

Finally, neural representations themselves are not as clearly categorized as ego- or allocentric as was once thought. While place cells have been thought to be largely invariant to head direction of the animal, which is coded by a different set of head direction cells [27], some recent studies in virtual reality and physical environment have also shown that a majority of CA1 neurons are jointly modulated by spatial location and head direction [28, 29]. A potentially more fruitful approach might be to acknowledge that for neural representations and strategies, the distinction of allo- and egocentric is a gradual, and not a categorical property.

Here we begin to address some of the issues discussed above with our computational model that uses deep reinforcement learning to solve two distinct behavioral tasks: guidance and aiming [30] (Fig 1A) using allocentric and egocentric navigation strategies (Fig 1B). In guidance, the agent must navigate to a fixed unmarked goal in its environment using stable landmarks to guide its actions. In aiming, the goal is marked by a visible cue, which the agent approaches during navigation. Our hypothesis is that these two tasks should evoke very different navigation strategies and show a distinct relationship to the spatial representations that emerge in the agent.

## Materials and methods

### Tasks and simulation environments

All simulations were carried out in virtual environments using the CoBeL-RL (Closed-loop simulator of complex behavior and learning based on reinforcement learning and deep neural networks) modeling framework [31]. In the guidance task, the agent started from a random location and had to navigate to a fixed, unmarked goal location within a square environment of size 2.75m × 2.75m, unless otherwise specified. Distinct colors on the walls acted as distal landmarks to aid in localization. In the aiming task, we used a red cylinder as a salient cue to which the agent had to navigate within the same square environment used in the guidance task. The goal location and the cue were moved to a new random location every 10 trials, and the agent started every trial at a randomly chosen position. In both tasks, the agent received a positive reward of +1 upon successfully reaching the goal location. In order to encourage the agent to take shorter paths, we also assigned a negative reward of -1 for each step taken. When the agent attempted to walk into a wall or corner, it simply remained in place and the action was penalized with a -1 negative reward.

For both tasks, we trained the agent for a total of 4000 trials unless otherwise stated. A trial terminated when the agent reached the goal location, or after 100 time-steps, whichever

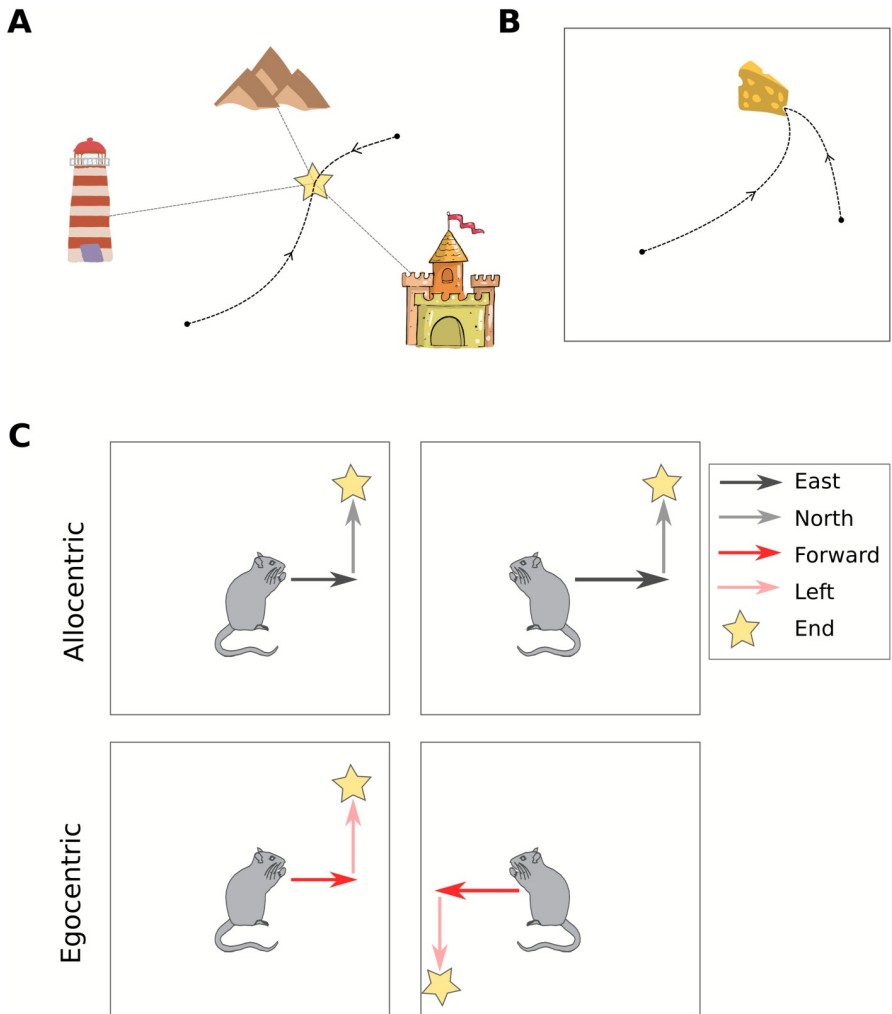

**Fig 1. Tasks and navigation strategies. A**, Schematic of the guidance task. The goal is unmarked, but fixed in space and can be identified in relation to stable landmarks. **B**, Schematic of the aiming task. The goal position changes frequently and is marked by a visual cue. **C**, Navigation strategies. Following the same sequence of actions using an allocentric navigation strategy leads to the same final location irrespective of the starting heading direction. However, when using an egocentric navigation strategy the same action sequence will lead to different final locations depending on the initial heading direction.

occurred first. The agent received as naturalistic input 48 × 12 pixel RGB images generated by the virtual environment and had a 240 degree field of view in all simulations (Fig 2A).

## Deep reinforcement learning model based on naturalistic visual inputs

We frame spatial navigation as a reinforcement learning problem (for a review of this approach, see [32]. In reinforcement learning, the agent must learn by trial and error while interacting with its environment. At each time step $t$, the agent finds itself in a current state $s_t$ and chooses an action $a_t$. We use an $\varepsilon$-greedy policy to select the action and handle the exploration-exploitation trade-off. That is, at each time step, the agent chooses a random action with probability $\epsilon$ ($\epsilon = 0.3$ in all simulations), and otherwise chooses the action $a_t$ that yields the highest value to the agent. The action causes feedback from the environment in the form of a scalar reward $r_t$ to the agent and a transition to the next state $s_{t+1}$.

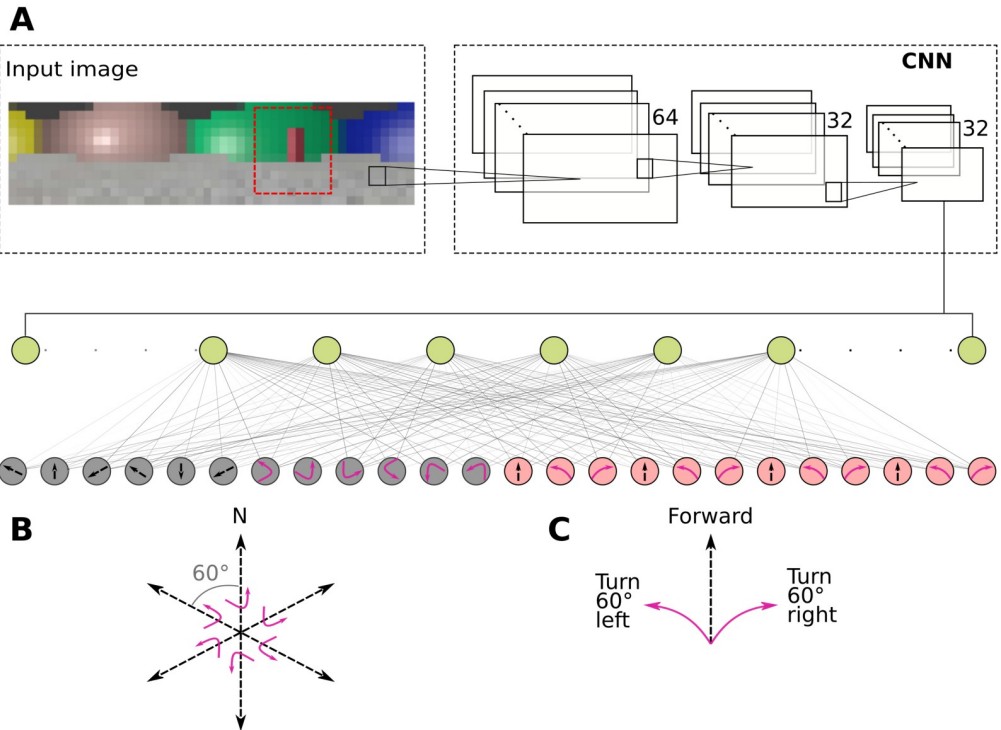

**Fig 2. Schematic of the Deep-Q network. A**, Visual inputs from the environment are processed by the CNN layers and then to a fully connected layer with 50 units (green circles) from where the spatial representations are analysed and classified. The output layer consists of allocentric (grey circles) and egocentric (red circles) action units in the full model, and only ego- or allocentric units in the constrained models. **B**, Schematic of the allocentric action space. **C**, Schematic of the egocentric action space.

In reinforcement learning, the agent adapts its behavior to maximize the future cumulative discounted reward. To this end, we adopted the Q-learning algorithm [33], where the agent learns a state-value function $Q(s, a)$, which is updated according to the following rule.

$$Q(s_t, a_t) \leftarrow Q(s_t, a_t) + \alpha[r_t + \gamma \max_a Q(s_{t+1}, a) - Q(s_t, a_t)] \tag{1}$$

Traditional Q-learning maintains a table of state-action pairs along with the associated Q-value. This however requires making some assumptions about how the environmental states are encoded. In order to keep these assumptions about the input representations to a minimum, we used naturalistic visual inputs as described above. Due to the large size of the state space, it was necessary to use an artificial neural network as a function approximator (Fig 2A). This also enables us to analyze how the visual input is processed to give rise to spatial representations. The network consisted of a convolutional neural network (CNN) [34] that had three layers with 32, 64 and 64 filters respectively. Each filter had a size of $5 \times 5$. The CNN was followed by a fully connected layer with 50 units. The output layer consisted of action units, where each unit corresponded to an available action. We used rectified linear unit [35] as activation function throughout the network, except for the output layer, which used a linear activation function.

We used the Deep Q-Network (DQN) algorithm [36] to train the network. We applied double-Q [37] and dueling network [38] modifications to stabilize the training. As with most deep reinforcement learning algorithms, we used experience replay with a buffer size of 3000 experiences to train the network. For all simulations we used a learning rate of $\alpha = 0.001$ and

discounting rate of $\gamma = 0.99$. We also applied a dropout rate of 0.35 to the fully connected layer in all simulations, unless otherwise specified.

## Navigation strategies and action spaces

We used allo- and egocentric navigation strategies to solve both tasks. The strategies were realized in practice using the agent's action space. Because the action space defines the reference frame in which navigation takes place, in our model, we view the strategy as being identical to the action space used. While it is in principle possible to define strategy use differently, we believe that equating it to the action space allows for a straightforward and direct interpretation of the results, as well as a clear separation of the strategy from the spatial representations that underpin it.

The action space was the set of all actions available to the agent. In the allocentric action space, the agent could either move by a fixed distance in one of six allocentric directions, or rotate in place to one of six head directions defined in allocentric coordinates (e.g. face 90 degrees due North). Translations and heading directions were constrained such that the agent moves on a hexagonal grid (Fig 2B). That means that the six different movement and heading directions are aligned with the main axes of the grid, each 60 degrees apart, and one movement takes the agent from current node to a neighboring node on the graph. Furthermore, the goal is always placed on a node of the grid to ensure that it can always be reached by the agent despite its motion. By contrast, in the egocentric action space, the agent could choose to move forward from its current position by a fixed distance or turn in place by a fixed angle of 60 degrees either to the left or right of the current heading direction, similarly corresponding to navigating on the same hexagonal grid (Fig 2C). The egocentric agent must thus successively make three left or right turns to change its orientation by 180 degrees and turn around completely.

In the full model, when both action spaces are available to the agent, there would be a mismatch in the number of physical actions between the allocentric (12 actions) and egocentric strategy (3 actions). To avoid potential biases due to this mismatch, we included four copies of each egocentric action in the full model to equalize the number of available actions between the two action spaces. During training, credit was assigned only to the particular copy of the action that was chosen, not to all four copies of that action. We assessed whether this design choice made learning in the egocentric agent noisier by comparing learning curves in both tasks using either single or multiple copies of each action and found no discernible effects (S1 Fig). In the purely allocentric and egocentric variants of the model, where the action spaces do not compete, only one copy of each action was available to the agent.

## Analysis of spatial representations in the network

Once the agent had learned the tasks, we computed spatial activity maps for each unit in the layer before the action selection layer (Fig 2A). The spatial activity maps were computed by partitioning the environment into a $25 \times 25$ grid and recording the activation of the units at each point on the grid and applying a Gaussian filter with a standard deviation of 2. This was done for six different head directions. For the aiming task, we also computed activity maps by placing the agent at the center of the environment and moving the goal cue on a $25 \times 25$ grid. Six separate maps were computed this way for six different head directions. This enabled us to identify the preferred field of activity of a unit relative to the position of the cue, much like a receptive field.

Based on the spatial activity maps, we classified the units into place cells, egocentric vector cells, head-direction modulated cells, and view-selective cells. To do so, we first identified

whether the spatial activity map of a unit had a localized field (active in <50% of the environment) by finding the peak of activation and marking the area where the activation fell to <15% of the peak. If the unit had a localized firing field whose field center did not change location (within 5% of arena size) with head direction, it was identified as a place cell. If the unit had a localized field whose center was always at a fixed distance and direction from where the agent was facing, it was classified as an egocentric vector cell. We also identified direction-dependent firing fields, which had localized firing fields that were modulated by head direction, as well as view-selective units that were only active in a single head direction or a subset of head directions, corresponding to a particular view seen by the agent. If the cell showed neither localized nor directional firing fields, it was classified as "other" (S2 Fig).

For both aiming and guidance, we also tested the agent's ability to generalize by using training and test sets. For guidance, the agent was trained using a subset of all possible start locations, and had to navigate to the goal from novel start locations in the test phase. In aiming, the training set was composed of a subset of goal locations. The agent was then tested on novel goal locations to gauge generalization. The training set consisted of 80% of all possible start locations for guidance, and 80% of all possible cued goal locations for aiming. The remaining 20% unseen start and goal locations were used as test sets for guidance and aiming respectively. The assignments were made randomly for each agent before training started.

A further set of environmental manipulations was designed to increase the task difficulty specifically for the non-preferred strategy. For guidance, we used three different arena sizes: the arena used in all regular simulations (2.75m × 2.75m) and two larger environments of size 4m × 4m and 5.5m × 5.5m. We hypothesized that the larger the environment is, the larger the difference in difficulty of the task would be for the egocentric strategy compared to the allocentric strategy, because the views would differ more from one another in larger environments.

In the aiming experiment, we gradually removed spatial information, which makes it harder to navigate using the allocentric strategy. In the high spatial information case which was used in all regular simulations, the environment had walls and point lights, which are both sources of spatial information. In the medium spatial information case, we removed walls, leaving only point lights as sources of spatial information. Finally, in the low spatial information case, the environment had no walls or point lights, and was instead only lit by a uniform distant light source.

Next, we used network manipulations to identify which units were critical to the performance of the agent. In a test phase, we first disrupted one unit at a time selectively by injecting noise into that unit. The noise was drawn from a normal distribution, scaled by the maximum activation in the layer. We then tested the agent for 25 trials on its ability to find the goal. We also disrupted populations of units classified by the type of spatial representation in a similar manner.

## Results

We defined two behavioral tasks (guidance and aiming) and two action spaces corresponding to allocentric and egocentric navigation strategies. Our deep reinforcement learning model learned to solve both tasks using only high-dimensional naturalistic visual inputs.

### Choice of navigation strategy emerges from task demand

We first trained an agent with both action spaces, i.e. navigation strategies, without any constraints to determine whether a preferred strategy emerged naturally for each task. The agent learned both tasks efficiently (Fig 3A). The agent had the option of acting in accordance with

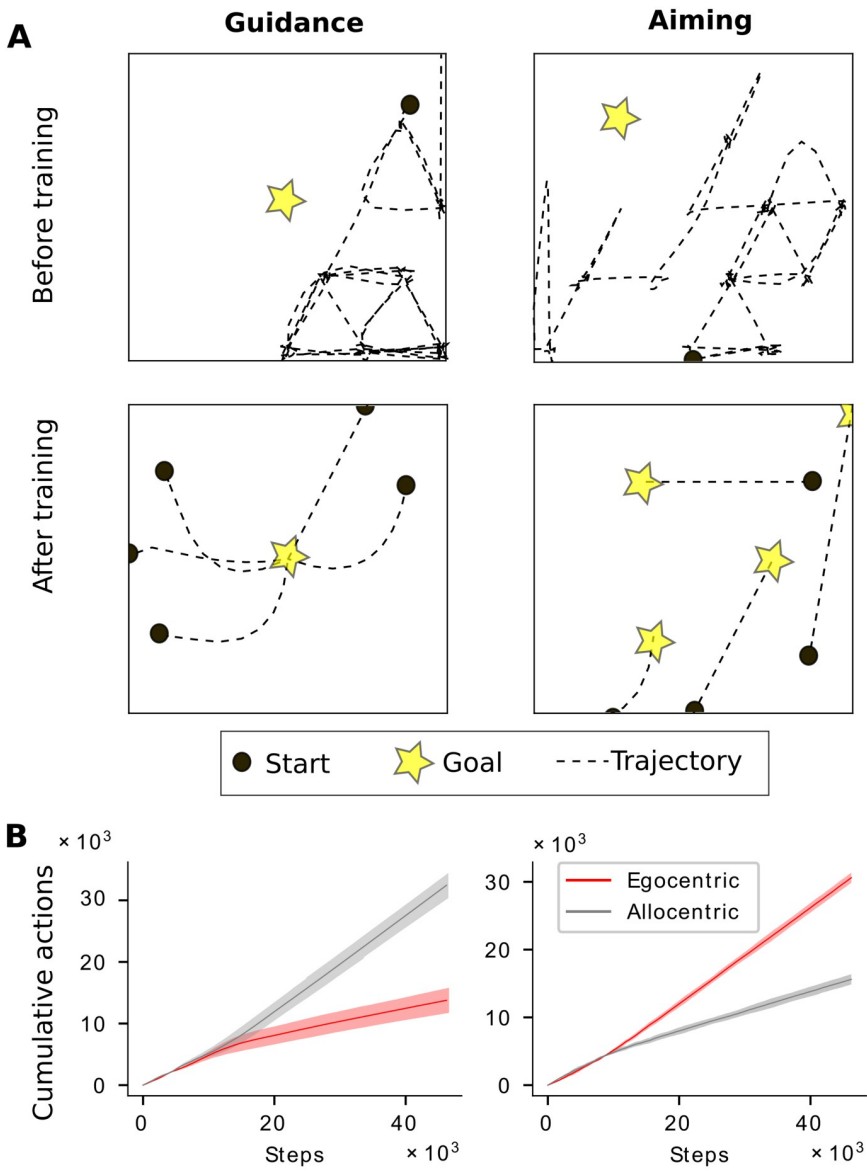

**Fig 3. Task demands drive choice of navigation strategy and emerging spatial representations in the full model. A**
Example trajectories before and after learning in the full model for guidance and aiming. At the end of training, agents
have learned to take direct paths to the goal. Trajectories are smoothed for visualization. **B**, A clear preference for allo-
and egocentric actions emerge for guidance and aiming in the full model, respectively. Dark lines represent means over
15 simulations. Shaded regions represent the standard deviations.

an egocentric or allocentric strategy, i.e., the agent could choose to use only egocentric or allo-
centric actions, respectively. Alternatively, the agent could choose to use a mixture of actions.
We found that, when solving the aiming task, the agent preferentially chose actions in an ego-
centric reference frame, whereas when solving the guidance task, the agent preferentially chose
actions in an allocentric reference frame (Fig 3B). As mentioned in the Introduction, an allo-
centric reference frame is intuitively more useful for guidance, because it involves navigating
to a fixed location in space based on stable landmarks. In this setting, every configuration of
landmarks can be assigned to an allocentric location within the environment. The egocentric
strategy, on the other hand, is better suited to aiming due to the lack of a relationship of stable

landmarks to the goal position, which is instead marked by a cue whose position changes from session to session. In this situation, a parsimonious strategy would be to base navigation decisions on the position of the cue with respect to oneself [39]. Our result suggests that for a given spatial navigation task, there is a preferred navigation strategy that is better matched to the task's computational demands.

## Spatial representations are modulated by task in addition to navigation strategy

We next examined the spatial representations underlying the strategy in each task by plotting spatial activation maps for each unit in the layer preceding the output layer. We found that stereotypical spatial representations (place-like and vectorial) were present in both guidance and aiming. When the agent solved guidance, place cell-like representations emerged (Fig 4A), and covered the whole environment (Fig 4B). When the agent solved aiming, egocentric vector-like representations emerged (Fig 4C). In both tasks, we also found a number of head direction modulated cells, consistent with experimental results [18, 28, 29, 40]. We also found that the place cells tended to cluster around the goal location (Fig 4B) similar to what has been observed experimentally [22, 23]. In addition to place and vector cells, we also observed a number of other types of responses including head-direction modulated cells, and those that were active only in a subset of viewing angles (Fig 5) in the full model.

Having established that there is a preferred navigation strategy for each task and having examined the emerging spatial representations, we next analyzed how tight the relationship between task, navigation strategy, and spatial representation is. To do so, we constrained our model to use either the preferred or nonpreferred strategy alone to learn both tasks, and examined the differences in learning performance and the emerging spatial representations. In guidance, the learning curves were similar using either strategy, but there was a small difference in the asymptotic performance, with the agent performing slightly better using an allocentric strategy (Fig 6A). In aiming, the egocentric strategy learned the task much more quickly than the allocentric strategy (Fig 6A). It stands to reason that using the egocentric strategy is more efficient to learn aiming, because it requires learning a set of stimulus-response pairs for each specific view. By contrast, using an allocentric strategy requires combining different egocentric views corresponding to a single spatial location and using that knowledge to navigate [41]. This relative ease of using an egocentric strategy must be balanced with the appropriateness of using an egocentric or allocentric strategy for a given task, based on the type of information that is available in the environment. Indeed, allocentric memory deficits have been observed in subjects with mild cognitive impairment (MCI). For instance, Weniger et al [42] studied subjects' navigational abilities in a virtual reality park and maze, and found that participants with MCI were impaired in their ability to navigate in both environments as compared to controls. They also found that patients with MCI were strongly impaired in their performance on neuropsychological tests of allocentric memory. While the navigation deficits in the park seem to be associated with this impaired allocentric memory, the deficits in the maze are harder to interpret, since many participants were unable to even find a navigation strategy in this environment, again underscoring the importance of finding an appropriate strategy for a given task.

Next, we investigated the spatial representations that emerged during training in the constrained models. If they were modulated by the navigation strategy alone, the representations should remain relatively stable across tasks, if the same strategy was used to solve them. If, on the other hand, spatial representations were modulated by task alone, they should be similar for different solutions of the task, regardless of the strategy. We found that navigation strategy

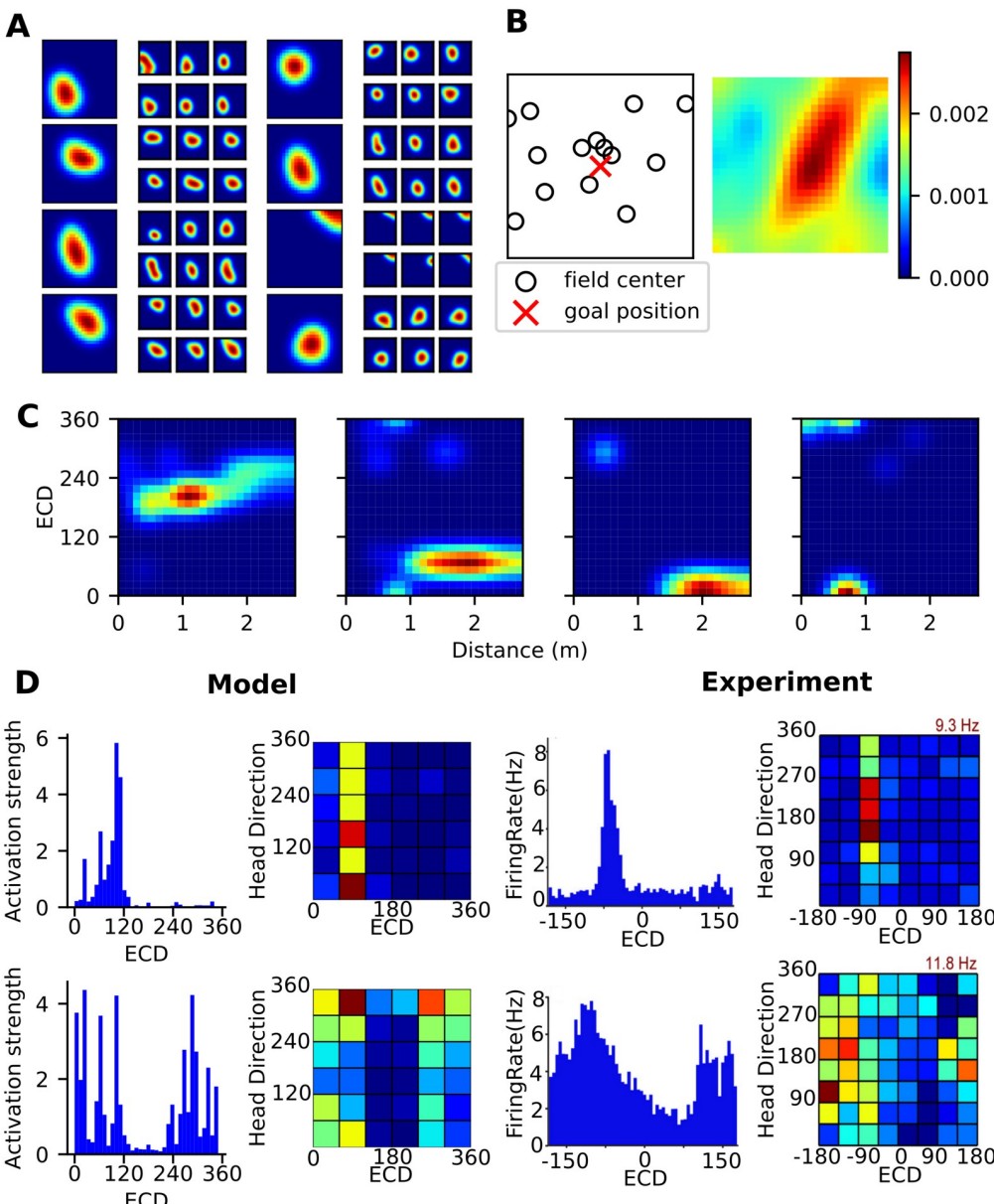

**Fig 4. Stereotypical spatial representations emerging in the guidance and aiming tasks. A**, Sample place cell-like responses in guidance using an allocentric strategy. Larger firing field maps show head-direction averaged responses of the unit, and the six adjacent smaller maps show the corresponding response for individual head directions spaced 60 degrees apart. **B**, Place fields cover the entire arena, but tend to cluster around the goal location. Left, location of place field centers in the arena. Right, corresponding normalized sum of firing rate maps of all place-like cells. The entropy of the distribution corresponding to the place cell coverage is $H = 9.035$ (for a uniform distribution $H = 9.288$) **C**, Sample vector-like representations in the aiming task using an egocentric strategy from our model, showing tuning to both the egocentric direction (ECD) and distance to the cue that marks the goal. **D**, Left, Two sample vector-like representations in the aiming task using an egocentric strategy from our model. Right, experimentally observed representations of ECD in mice navigating to cue lights in the environment (reproduced with permission from [18]).

alone did not account for spatial representations (Fig 6B). When using an allocentric strategy to learn guidance, the agent developed stereotypical place cell-like responses, however, the same strategy lead to few allocentric responses when learning aiming. Likewise, when using an egocentric strategy to learn aiming, egocentric vector-like responses constitute the majority of

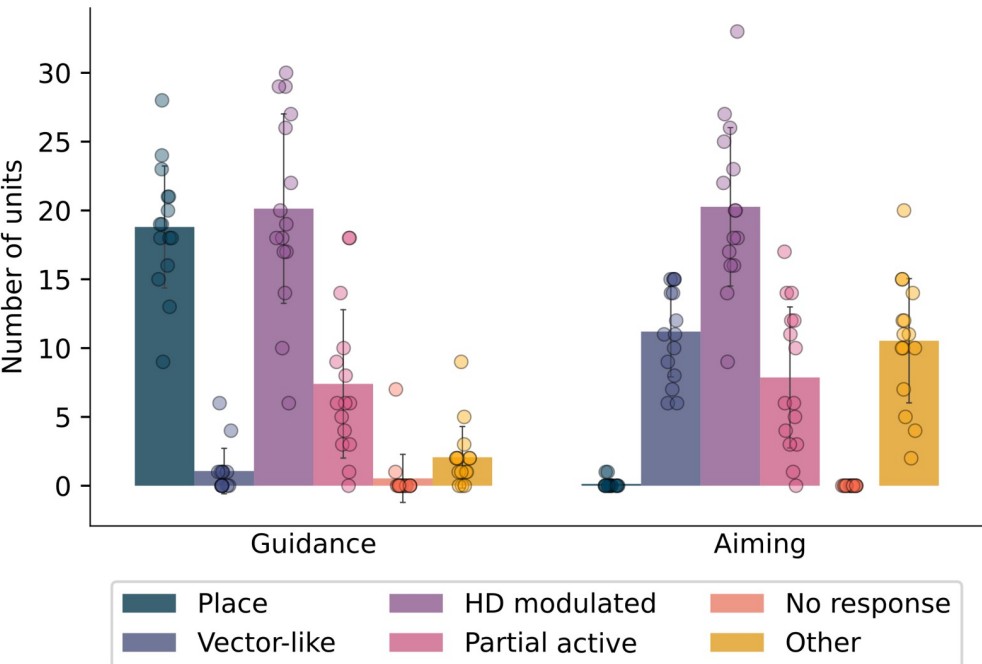

**Fig 5. Classification of units in the network.** Spatial representations emerging in guidance and aiming show clear differences commensurate with the preferred navigation strategy. Bars represent means over 15 simulations, the outcome of each is represented by a circle. Error bars represent the standard deviation.

spatial representations, when using the same strategy, but vector-like responses did not not persist when learning guidance. Conversely, task alone did not account for spatial representations either (Fig 6B). Learning aiming with an allocentric strategy yielded far fewer egocentric responses than with an egocentric strategy, and solving guidance with an egocentric strategy yielded no place-cell-like responses. Hence, our simulations revealed that neither navigation strategy nor task demands alone determine the evolution of spatial representations, but rather a combination of the two (Fig 6B).

Another significant aspect of allocentric place-cell-like responses in the model is their relatively slow emergence during learning in comparison to egocentric responses. We hypothesize that this is because different egocentric views must be associated with the same spatial location in order for an allocentric representation to be generated and that this process requires a larger number of learning trials than learning view-dependent behaviors. We also see a significant number of head-direction modulated responses and view-selective responses across all task-strategy combinations, consistent with experimental recordings [28, 29, 40].

## Spatial representations contribute to generalization of learned behavior

If the agent is able to solve the guidance task without allocentric place-like representations and even the learning speeds are similar (Fig 6A), why do the allocentric strategy and representations emerge preferentially for the guidance task in the full model? It seems unlikely that these effects are driven by the slightly better asymptotic performance of the allocentric strategy. Instead, we hypothesize that the agent using the egocentric strategy essentially memorizes a sequence of actions based on sensory snapshots for different egocentric perspectives. This is inefficient for navigating in a stable, unchanging environment and does not lend itself well to generalization. We therefore test generalization in both guidance and aiming. In order to test

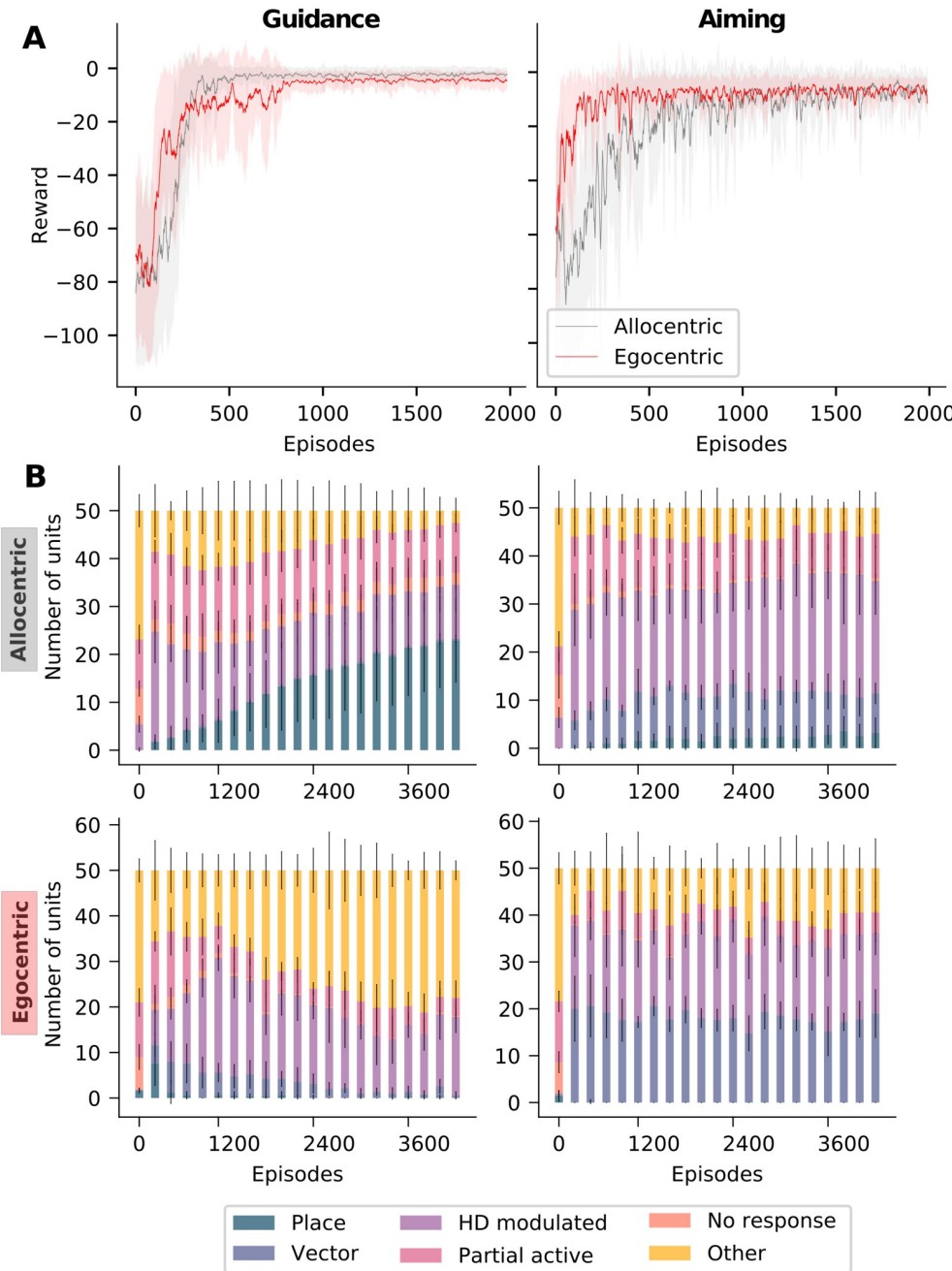

**Fig 6. Evolution of learning and spatial representations. A**, Learning curves for guidance and aiming. Curves show average reward at each learning episode over *n* = 10 runs. The shaded area represents the standard deviation. In guidance, the speed of learning for both strategies is similar, but the asymptotic performance is slightly better for the allocentric strategy. In aiming, the learning is accelerated when using an egocentric strategy as compared to using the allocentric strategy. **B**, Evolution of spatial representations with learning for all four task-strategy combinations.

generalization in guidance, the agent was trained using a subset of all possible start locations, and had to navigate to the same unmarked goal from novel start locations in the test phase. For testing generalization in aiming, the agent was trained on a subset of cued goal locations and had to navigate to novel goal locations in the test phase. Furthermore, we hypothesize the

disadvantage of the egocentric strategy should become more pronounced as the size of the arena increases, since action sequences that have to be memorized become longer, and the chances of reaching the goal purely by chance are reduced. On the other hand, since the allocentric strategy leads to learning a more general solution and allow the agent to navigate more flexibly from different starting points to the goal in the guidance task. Interestingly, learning during training is not affected much by arena size for either strategy (Fig 7A). However, the generalization performance for the egocentric strategy indeed deteriorates more than for the allocentric strategy as the arena size increases (Fig 7B). Our results suggest that learning alone is not a reliable marker for strategy use, and examining generalization performance might be necessary to effectively determine if the strategy being used is allocentric or egocentric.

Conversely, we hypothesize that in aiming the allocentric strategy can memorize the action sequence to a visual cue based on location-response associations. Therefore, we expected that if aiming was tested with novel goal locations that had not been presented during training, generalization would be inferior for the allocentric strategy compared the egocentric strategy. Furthermore, if we progressively removed distal spatial information, making it more difficult for the agent to localize itself using landmarks, the allocentric strategy would have a harder time learning the task and generalize even less. Indeed, learning and generalization performance show a sharp decline in the allocentric strategy as spatial information is removed in our computational simulations (Fig 7B and 7D). By contrast, since the egocentric strategy learns a more general approach based on vector-based representations of the goal, learning performance is independent of distal spatial information (Fig 7C). Strikingly, the egocentric strategy actually generalizes better when less spatial information is present in the environment (Fig 7D). This could be due to less confounding information in the visual input leading to the emergence of better representations of the informative cue.

Finally, another method for testing the generalization hypothesis is to study how regularization, in the form of dropout, affects the proportion of place-cell-like responses in the network in the guidance task. In machine learning, dropout is a method where a certain fraction of units, the dropout rate, is chosen at random and inactivated during a learning trial. That is, the dropped units neither participate in generating the network output, nor in the subsequent error-driven learning. Increasing the dropout rate forces the network to look for more general solutions. In our model of guidance, we found that it increases the number of place-cell-like representations in our simulations, the most were found in a network with 50% dropout (Fig 7E). Learning became unstable at higher dropout rates. This is similar to findings from other computational models that show that regularization is crucial to the emergence of stereotypical spatial representations [43, 44]. We however do not observe a similar effect for vector-like representations in aiming, where the proportion of these units remains stable with changing dropout. We hypothesize that this is due to a ceiling effect caused by the fact that a large number of vector-like representations are already present even at zero dropout rates, presumably because these representations are critical to solving aiming with an egocentric action space.

In conclusion, our results confirm that using the preferred strategy for each task results in more general solutions to the task, which have greater flexibility when task parameters change.

## Causal influence of spatial representations on navigation behavior

Having gained a better understanding of the important role that place-like cells and egocentric vector cells play in solving guidance and aiming, respectively, we next set out to study the causal role that the different cell types play in driving behavior. Specifically, we asked whether the stereotypical spatial representations indeed play a special role in the navigation task, and what, if any role the other types of spatial representations play. To this end, we selectively

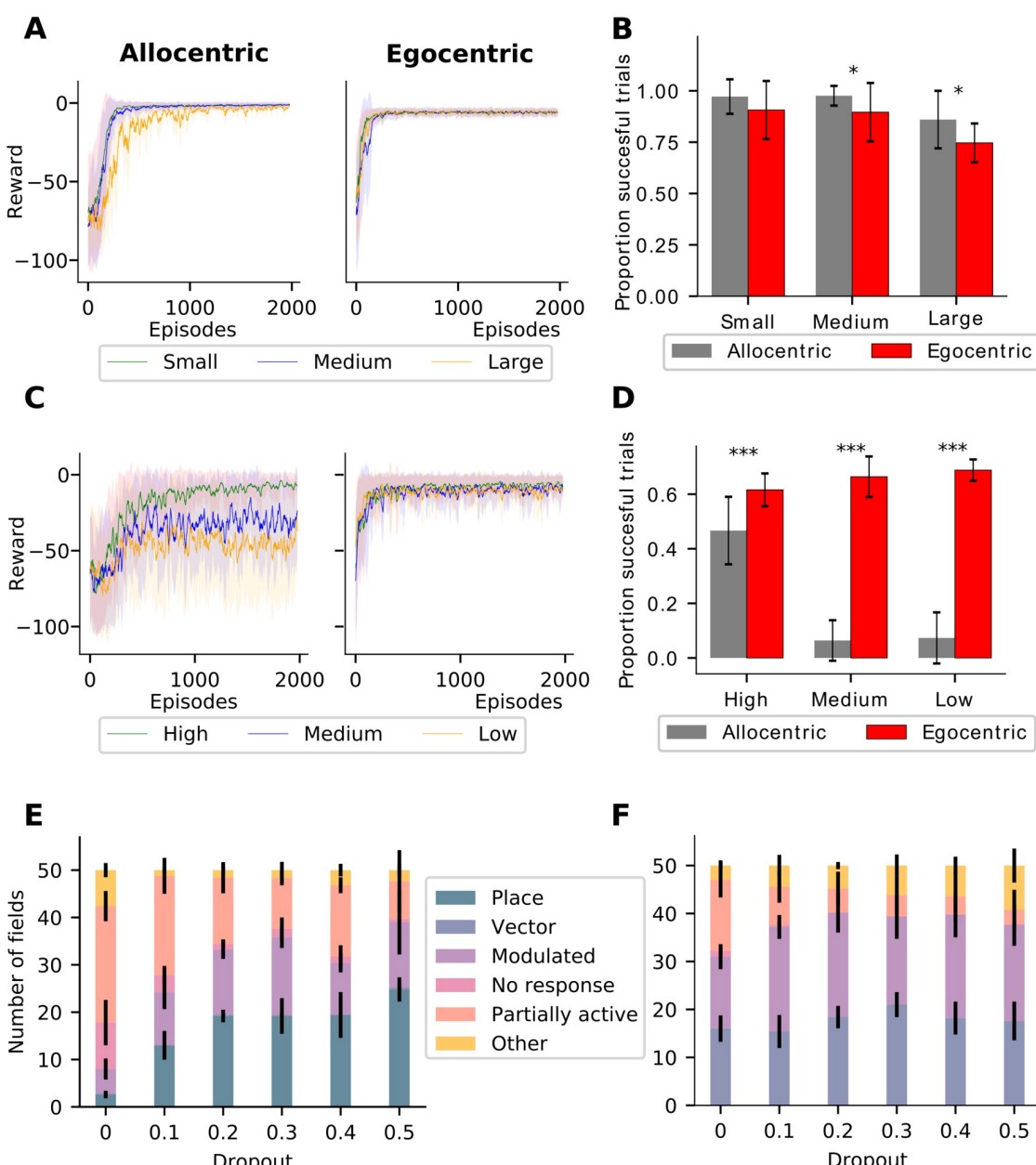

**Fig 7. Preferred strategy use leads to superior task generalization. A**, Learning curves for small, medium and large environments in guidance. **B**, Generalization performance of agent in guidance (successful trials during test after being trained from different start locations) for small, medium and large environments. Egocentric strategy shows worse generalization performance in all environment sizes. **C** Learning curves for aiming with different amounts of spatial information in the environment using allocentric and egocentric strategies. **D**, Generalization performance of agent in aiming (successful trials during test after being trained to navigate to different cued goal locations) for environments with different amounts of spatial information. **E**, Effect of regularization in the form of dropout for guidance task, which encourages the network to look for more general solutions. In guidance, the number of allocentric place-cell-like responses increases with the degree of dropout applied to the network. **F**, Effect of dropout for aiming task. An increase of vector representations with dropout is not observed, which could be due to the presence of a ceiling effect.

disrupted different types of units in the network by injecting varying levels of noise. We did this in one of two ways. First, we injected noise into one unit at a time, and measured the performance of the agent in a test phase. Second, we injected noise into entire populations of the same kind of units, for instance, we injected noise into all place-cell-like units at the same time

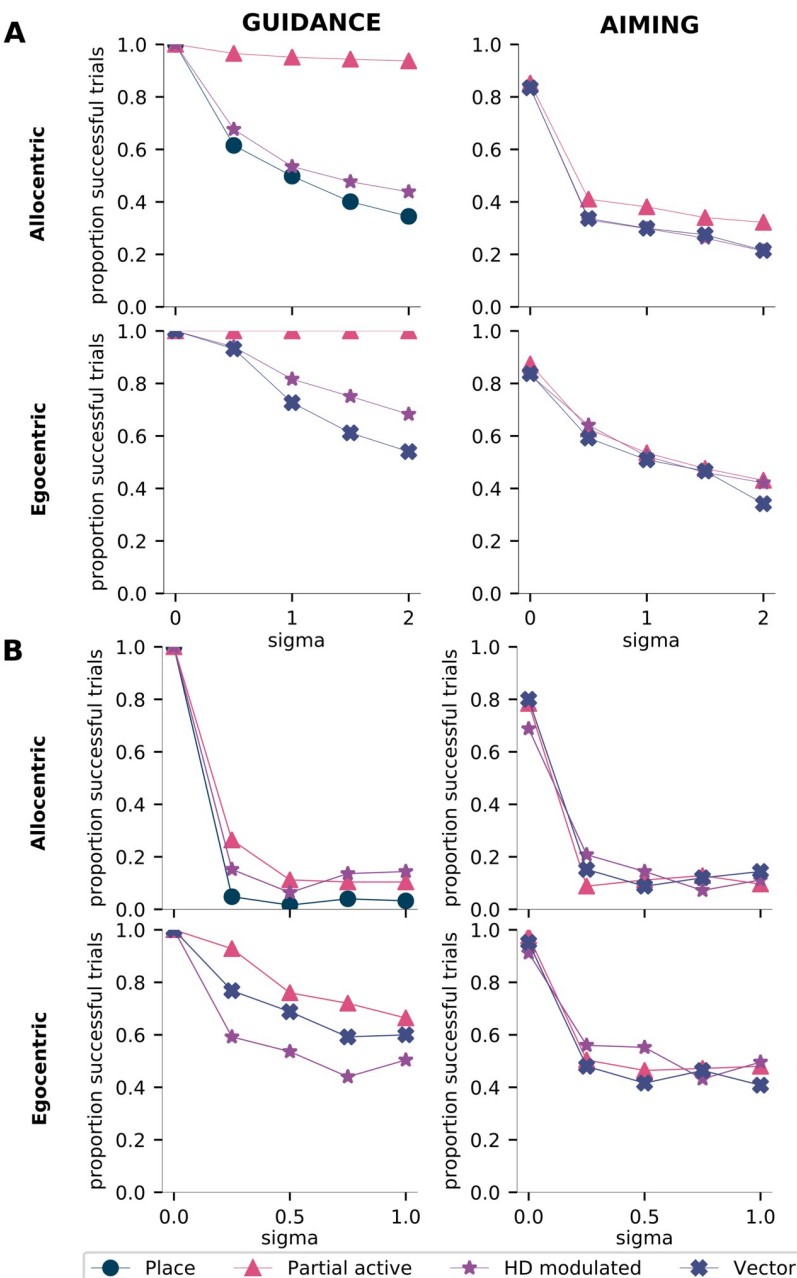

**Fig 8. Different types of spatial representations reveal certain types play a special role. A**, Effect of injecting noise into single units of different types. **B**, Effect of injecting noise into populations of units of the same type. Horizontal axes represent the standard deviation of the normal distribution from which noise signals are drawn, while the vertical axes indicate the proportion of successful trials in a test phase.

(for details, see Methods). As expected, the more noise is introduced into the network the more the performance of the agent deteriorates.

In allocentric guidance, disrupting place-cell-like units, either individually or as a population, has the greatest effect on the agent's performance (Fig 8). Thus, each individual place cell is important for allocentric navigation and serves as a basis for the agent to localize itself. In aiming, disrupting single units regardless of which type has a catastrophic effect on

performance (Fig 8A). So, individual vector cells are crucial for performance, but not more so than other unit types. This effect could be due to an overlap in function between vector cells, head direction modulated cells, and view selective cells, as they are all egocentric in nature. A similar trend is observed while disrupting populations in aiming (Fig 8B).

In conclusion, while place-like units appear to play a relatively more important role in task performance than other unit types in guidance, they do not support behavior in isolation, and multiple unit types play an important role in task performance in aiming.

## Discussion

In our computational model, we looked at navigation through the lens of deep reinforcement learning and studied the interplay of task, navigation strategy, and spatial representations. We defined two ethologically valid tasks for mammals, namely, guidance and aiming, and used our model to learn them using allocentric and egocentric strategies. This allowed us to establish a preferred strategy for each task, which aligns with experimental results [45]. Strikingly, we find that when constrained to do so, our model is able to learn either task using the nonpreferred strategy as well. We computed spatial activation maps for the units in the model and found that our model develops both allocentric place-cell-like representations and egocentric vector like representations. We found that the emergence of specific types of representations depends on the task as well as the navigation strategy. We found that although the model can learn both tasks using the nonpreferred strategy, using the preferred strategy confers the benefit of better generalization. Our model predicts that in tasks where it is possible to use stable landmarks to navigate, animals that primarily use an allocentric strategy would develop more place cells as compared to animals that base their actions on an egocentric strategy, and would also be able to generalize better.

### Relationship to coordinate transformations in the brain

To use an allocentric cognitive map, the brain must implement coordinate transformations, because perceptual inputs are inherently egocentric, changing with eye, head and body position, and motor outputs are also egocentric in nature [46, 47]. Hence, these egocentric inputs must first be transformed into the allocentric reference frame that the map employs, and then back to egocentric coordinates for motor commands. In our model, the visual inputs are indeed egocentric in nature, albeit somewhat simplified compared to navigating animals, because we do not model independent eye and head movements. In the cases where allocentric representations emerge in the model, there is an implicit coordinate transformation on the input side, because the model aggregates the different egocentric views at a single location into an allocentric representation of that location. By contrast, there are no coordinate transformations on the output pathway of our model. Even though we call the outputs of our model "actions", they are not motor commands, but rather higher-level navigation decisions, which have to be converted into motor commands by a downstream system. When using the allocentric "action space", such a downstream system would have to explicitly or implicitly implement coordinate transformations. However, even the egocentric actions outputted by our model have to be translated by a downstream system into motor commands. This is because ultimately all navigation decisions have to be transformed into the coordinates of the relevant muscles for movement. This means that the same higher level navigation decisions may require very different coordinate transformations based on the current posture and navigation method ("turn left" on a bicycle vs. while walking).

## Implications for experimentally studying spatial navigation strategies

Many studies have shown that allocentric representations and navigation strategy are more susceptible to disruption by aging and disease compared to egocentric strategy use [3–6]. For example, older humans perform worse in spatial cognition tasks in real world [48, 49] and virtual environments [50, 51]. Decline in allocentric strategy use in preclinical Alzheimer's disease [52, 53], as well as in other pathologies have also been observed [54–56]. Thus, properly distinguishing the use of these strategies in behavioral experiments is of paramount importance.

Why is the allocentric strategy more vulnerable to disruption by age and disease? It is possible that the allocentric strategy is cognitively more demanding, thus general cognitive decline associated with age is more likely to affect the allocentric strategy first. This is also reflected in our model in the fact that the agent generally takes longer to learn the allocentric strategy than the egocentric strategy.

It has also been suggested that traditional measures for determining the navigation strategy use are insufficient, and more fine-grained behavioral measures are needed [57]. Furthermore, seemingly allocentric tasks can also be solved using an egocentric view-matching strategy [11]. This is evident in our model as well, since we find that both tasks can be solved using either navigation strategy. Thus, as Rogers et al [57] suggest, a simple analysis of the behavior on the learned task might not be sufficient to determine strategy use. Our results suggest that in addition to developing more fine-grained behavioral analyses, testing for generalization performance can help in pinpointing strategy use. Thus, when designing behavioral experiments that aim to dissociate the use of allocentric and egocentric strategies, it is important to design probe trials that not only test for cue use, but also for generalization. In support of our prediction, Rinaldi et al [58] found that generalization and strategy use are correlated in a cross-maze task.

## Do allocentric and egocentric representations overlap in the brain?

Much debate surrounds whether allocentric and egocentric representations exist in parallel and separate neural systems, either interacting to support behavior (interacting model), independently supporting behavior without interaction (non-interacting model), or are largely overlapping, with purely allocentric and purely egocentric representations at opposite ends of a spectrum (overlapping model) (for a discussion of these models, see [39]. Another school of thought holds that the brain contains only egocentric reference frames, and that allocentric reference frames are simply a combination of several egocentric ones or the result of coordinate transformations [41].

Even though we analyzed and classified spatial representations in an abstract reinforcement model, we believe that these representations can be mapped to those in the brain. Nevertheless, we refrain from assigning concrete neural substrates to the different representations that arise in our model, because these representations are also distributed in the brain. Here, we highlight some neural representations that share similar properties with the representations emergent in our model. Firstly, place-like-units in our model correspond to the allocentric place cells that are found in the hippocampus [59]. Similar place-like units have also been found in the anterior cingulate cortex [60], subiculum [61], and entorhinal cortex [61, 62]. The head direction modulated cells in our model show localized firing fields, but are sensitive to head direction, like place cells in CA1 that are modulated by the head direction of the animal [28, 29]. Units in the hippocampus of Egyptian fruit bats and big brown bats show conjunctive sensitivity to location and head direction when the bats are engaged in crawling behavior [63], similar to the units that were classified as "partially active" in our model. These units also show

similarities to hippocampal cells in macaques which are selective to the monkey's heading direction and position [64]. Finally, the vector-like representations share properties with the egocentric cue direction cells discovered by [18] in the parietal cortex while engaged in an aiming task.

Our model supports a theory of overlapping neural representations that form a continuum, since purely allocentric or purely egocentric representations never emerge in isolation in our model. Rather, both tasks invoke a mixture of the two types of representations, with the predominance of one or the other type mediated by strategy use. Indeed, we can artificially manipulate the tasks to tune the level of predominance, but most ethologically valid situations probably engage both representations and strategies to varying degrees.

Our conclusion might seem at odds with the long-standing suggestion that the hippocampus is the neural substrate of the allocentric cognitive map [9, 13, 17, 59]. However, there is now increasing evidence that shows that egocentric representations are also present in the hippocampus and other regions of the medial temporal lobe [18, 24, 29]. It is also worth pointing out that place cell recordings from the hippocampus on linear tracks and star mazes are predominantly unidirectional in nature [40, 65], which could imply an egocentric aspect to these representations as well. These findings are consistent with our results that suggest that the use of an egocentric reference frame and egocentric representations are ubiquitous compared to their allocentric counterparts. This is evidenced by the fact that our model can use the egocentric strategy to learn both tasks well, and egocentric representations arise in all task-strategy combinations, while allocentric representations emerge only in the particular combination of guidance and allocentric strategy use.

In a similar vein, our work also adds some perspective to the place vs. response literature (reviewed in [66]). Response based navigation is inherently egocentric in nature and typically develops with repeated experiences of the same route [67–70]. The dorsal striatum and specifically the caudate nucleus have been implicated in the development of these habitual responses [69]. While the tasks used in these dissociation studies indeed evoke the use of an allocentric or egocentric reference frame and can pinpoint the corresponding neural basis, there is an added layer of complexity because these reference frames are also being used at distinct levels—a more abstract, cognitive level associated with the place strategy, and the rote, habitual level associated with the response strategy. We believe that our study brings attention to the relatively understudied use of the egocentric reference frame in higher-level navigation planning and decisions, which might still depend on the hippocampus, thus leading to conflicting experimental evidence. This might explain why, for relatively simpler tasks, hippocampal animals are still able to use egocentric navigation strategies, but for more complex tasks, lesions to the hippocampus seem to affect both strategies [14]. Experimental results are also consistent with the suggestion that while regions at the lower, specialised levels of the navigation system are likely modular, such as the coordinate transformation system in the retrosplenial cortex [47] and habitual responses in the dorsal striatum [69], there may be less modularity in the higher, more general levels than previously assumed. We liken our model to an abstract higher-level navigation system which would need to be flexible and lend itself well to generalization, including being able to use and represent both frames of reference depending on the task.

## Reinforcement learning as a model for navigation

Reinforcement learning provides a natural framework for understanding navigation, since they are both goal-driven and must deal with uncertainty while balancing using known routes with exploring novel ones. Reinforcement learning also integrates sensory inputs from the

environment, behavior, and rewards into a single closed loop, and thereby models their interactions. Deep reinforcement learning makes it possible to process naturalistic visual inputs and analyze spatial representations that emerge in the network. As a result of these features many previous computational models have used reinforcement learning to model spatial navigation, e.g. previous models of path integration [43, 44].

Some models have mapped allocentric (or place-based) and egocentric (or response-based) strategies to model-based and model-free reinforcement learning algorithms, respectively [71, 72]. There is some experimental evidence supporting the mapping of these RL algorithms to the place vs. response dichotomy in rodents [73] and in humans, the hippocampus has been linked to model-based planning [74]. While these models approach approach the issue of how RL algorithms could be mapped on to a cognitive vs. behavioral navigation system, our model provides a complementary approach that looks at how different reference frames could be used at the cognitive level, and learns to navigate with either strategy using a simple model-free algorithm.

It is important to note that a major difference between deep RL agents and mammals learning to navigate is the typical time required to learn a navigation task. While rodents and other mammals are typically capable of learning these tasks within a few blocks of trials, RL agents, such as ours, require hundreds to thousands of training episodes depending on task complexity. Consequently, we do not necessarily propose that these strategies are learned or spatial representations emerge on a task-by-task basis in animals, but rather that these aspects are learned over longer time scales, whether developmental or evolutionary, and then tuned on a task-by-task basis.

Using model-free reinforcement learning helps provide an intuition about the computational demands of a task and the representations required to solve the task. For instance, why do stereotypical representations emerge in our model? Since the output action units in our model represent the Q-values of each of the actions, the preceding layer where we analyse the representations aids in the computation of this Q-value. Under an optimal policy, the Q-values for state-action pairs in guidance should depend on the action and the spatial location, but not on head direction. A simple way to achieve this outcome would be to insert the place-cell-like representations in the penultimate layer.

In conclusion, we have shown that the relationship between task, navigation strategy and spatial representations is more complex than previously thought. Both task and strategy played a critical role in shaping the spatial representations in our reinforcement learning model, and these representations determined the ability of our model to generalize to novel situations.

## Supporting information

**S1 Fig. Comparison of learning curves in the egocentric agent using multiple and single action copies in guidance and aiming task.**
(TIFF)

**S2 Fig. All spatial activity maps of units labeled "other" from a single run.**
(TIFF)

## Author Contributions

**Conceptualization:** Sen Cheng.

**Formal analysis:** Sandhiya Vijayabaskaran.

**Funding acquisition:** Sen Cheng.

**Investigation:** Sandhiya Vijayabaskaran.

**Project administration:** Sen Cheng.

**Software:** Sandhiya Vijayabaskaran.

**Supervision:** Sen Cheng.

**Visualization:** Sandhiya Vijayabaskaran.

**Writing – original draft:** Sandhiya Vijayabaskaran.

**Writing – review & editing:** Sen Cheng.

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
