## [Decision Letter · Decision Letter 0]

4 Aug 2022

Dear Ms. Vijayabaskaran,

Thank you very much for submitting your manuscript "Navigation task and action space drive the emergence of egocentric and allocentric spatial representations" for consideration at PLOS Computational Biology. As with all papers reviewed by the journal, your manuscript was reviewed by members of the editorial board and by several independent reviewers. The reviewers appreciated the attention to an important topic. Based on the reviews, we are likely to accept this manuscript for publication, providing that you modify the manuscript according to the review recommendations. In particular, the authors should provide some further details of how spatial representations in their model contribute to generalisation; and clarify some details of their methods and analyses, following the reviewers comments below.

Sincerely,

Daniel Bush

Associate Editor

PLOS Computational Biology

Samuel Gershman

Deputy Editor

PLOS Computational Biology

[LINK]

Reviewer's Responses to Questions

**Comments to the Authors:**

Reviewer #1: This study presents a model of learning allo- and egocentric navigation strategies using deep reinforcement learning (RL). The model takes the shape of a DQN that was allowed to take actions defined in egocentric space or actions defined in an allocentric space. The authors trained the model on two different navigational tasks, guidance and aiming, which are generally associated with allo- and egocentric navigation, respectively. Consistent with these intuitions, the model preferentially chooses allocentric actions during guidance and egocentric actions during aiming. In the final, fully connected layer of the network, place cell-like representations were found primarily during allocentric navigation, while egocentric vector representations were found more during egocentric navigation.

This study provides a valuable contribution to the literature on allocentric and egocentric navigation strategies. By using a deep network trained directly on visual inputs, the authors are able to show how both egocentric and allocentric strategies can be used to solve tasks that are associated with one or the other strategy, which is similar to experimental observations. Furthermore, their model allows ablation experiments showing the specific contributions of different cell types and highlights that differences in strategies might only be detected when looking at generalisation performance. There are thus many strengths to this manuscript. However, I think there are some important points to be addressed, see below.

1. I wonder how fairly the action spaces defined for the two models really reflect egocentric and allocentric “strategies”, rather than a difference in sensory and motor affordances. The allocentric actions comprise all six movement directions as well as turning around to any of the six directions, whereas the egocentric actions are only forward, left and right. The authors get around the imbalance in the number of actions by copying the egocentric actions four times but I think there are a couple of issues that remain with this modelling approach.

(a) A defining characteristic of allocentric navigation strategies is that allocentric directions still need to be transformed into the egocentric reference frame before movement. The authors mention in the discussion that we should interpret the actions in the model more as “higher-level navigation decisions” which still have to be translated into motor commands. I’m not so convinced by this argument, because if that is the case, shouldn’t that be the same for the egocentric actions too? Or are those lower-level decisions? Why can the egocentric agent not perform the action of turning around or walking backwards? These differences are not really the essence of egocentric and allocentric navigation but they will result in very different types of movement trajectories, where the egocentric agent needs to make a bigger circle.

(b) The allocentric agent can turn around on the spot and even walk backwards towards a goal. As far as I can see that strategy would be as good as turning around and walking forward to a goal, right?

(c) Have the authors tried making all movement directions, including the turning actions, available to the egocentric agent? Or alternatively, restricting the allocentric agent’s movement directions?

(d) It was unclear to me whether, when an egocentric action was chosen in the full model and a next state, reward pair was observed, all four action copies or only the chosen copy was assigned credit to. In the latter case, wouldn’t a different bias arise between the two strategies as a result of slower, noisier learning in the egocentric model? In the former case, would a bias still arise because backpropagated errors for egocentric actions have 4x the effect on weight changes in the network compared to allocentric actions?

(2) The authors rightly point out that there is a considerable overlap in regions involved in egocentric and allocentric navigation strategies. There is indeed evidence for both the striatum and the hippocampus being involved in egocentric and allocentric strategies. However, the authors fail to discuss several seminal papers that do show a clear dissociation between these regions.

(a) For example, Packard & McGaugh (1996) showed that hippocampal lesions affect (allocentric) place learning while lesions to the caudate affect learning of (egocentric) responses to cues. Pearce et al (1998) showed that hippocampal lesions selectively affect place learning in the water maze. It would be great if the authors could include a more extensive discussion of this literature.

(b) Do the authors think there is no modularity at all in egocentric and allocentric systems, or is there a softer form of modularity than previously assumed?

(c) What are the computational benefits and disadvantages of a modular architecture, and how does this relate to the present model?

(3) The authors’ hypotheses and simulations about generalisation were to me some of the most interesting in the paper. However I missed a few things in that section, which might be just an issue with exposition.

(a) It is unclear to me what exactly is shown in figures 7B&D. Is it the proportion of successful trials after having been trained from different starting points (B) or different goal locations (D)? This seems implied for 7B but is not very clear for 7D. Please clarify.

(b) If so, how were the novel goal locations chosen, and how were they distributed over the environment?

Fig 7E shows the satisfying result that increasing regularisation with dropout increases the place representations during guidance. Is this also the case for the vector-like representations during aiming? Why is this not shown? It would be nice to see the double dissociation.

Minor comments

- p5. For Q-learning, the reference should be Watkins & Dayan (1992).

- P312-314. Reference is made to fig 7C but I think this should be 7B. Also the error bars in 7B seem to overlap. Is the generalisation decay significant?

- What happens to the agents when they walk into a wall or corner?

- The authors mention in the discussion some previous models mapping allocentric and egocentric navigation strategies to model-based and model-free RL, respectively. Note that there is some experimental evidence for that mapping. For example, Kosaki et al. (2015) showed that place learning on the plus maze is devaluation sensitive (consistent with MB RL) while response learning is not. See also Vikbladh et al. (2019).

Reviewer #2: The authors presented a deep reinforcement model to study the emergence of spatial representations in allo- or ego-centric frame. Overall, the study is fine. However, the analysis is bit incomplete.

1) In Fig.4, the firing maps of the output units in the DQN are analyzed. For egocentric representations, it is likely that the distance and directions to the goal are encoded in the activities of the units. Have you tried to show the firing rate maps as a function of the distance and orientation to the goal in the egocentric frame? From fig 5, it can be seen that the percentage of other cells is quite high (~10%) in aiming task. It would be interesting to see if these cells actually encode some information.

2) In simulation, the distance the agent moves within one step is not given clearly. If this distance is large, and due to the discretization, the agent may not reach the goal. The authors should clarify this point.

3) From Fig. 7A,C, it seems that egocentric representation is more robust than the allocentric representation. The description in the text does not match the figure, since in the text, the authors wrote:”However, the generalization performance for the egocentric strategy indeed deteriorates more than for the allocentric strategy as the arena size increases (Fig. 7C).”

4) In order to test generalization, the authors use novel stating positions. The change introduced is somewhat small. It would be interesting to see the generalization performance with more drastic changes: train the network in small sized environment, and test the performance in large and medium-sized environments.

**Have the authors made all data and (if applicable) computational code underlying the findings in their manuscript fully available?**

Reviewer #1: Yes

Reviewer #2: Yes

PLOS authors have the option to publish the peer review history of their article (what does this mean?). If published, this will include your full peer review and any attached files.

Reviewer #1: No

Reviewer #2: **Yes: **Bailu Si

Figure Files:

Data Requirements:

Reproducibility:

References:

---

## [Decision Letter · Decision Letter 1]

5 Oct 2022

Dear Ms. Vijayabaskaran,

Thank you very much for submitting your manuscript "Navigation task and action space drive the emergence of egocentric and allocentric spatial representations" for consideration at PLOS Computational Biology. As with all papers reviewed by the journal, your manuscript was reviewed by members of the editorial board and by several independent reviewers. The reviewers appreciated the attention to an important topic. Based on the reviews, we are likely to accept this manuscript for publication, providing that you modify the manuscript according to a couple of minor editorial recommendations, listed below. Specifically:

[1] In the Introduction, "However, there three major issues with this view" should be "However, there are three major issues with this view"

[2] In the Results, where you state that "...studies have shown that the use of the allocentric strategy is associated with general cognitive decline [Weniger et al., 2011]", you may wish to provide a few more details of the cited study so that the implications are clearer to the reader

[3] When discussing whether RL may be a good model for navigation, it would seem highly relevant to mention that the length of training used here (and in other RL models of navigation) is typically much greater than that required by rodents and other mammals completing similar tasks

Sincerely,

Daniel Bush

Academic Editor

PLOS Computational Biology

Samuel Gershman

Section Editor

PLOS Computational Biology

Reviewer's Responses to Questions

**Comments to the Authors:**

Reviewer #1: The authors have adequately addressed all my previous comments.

**Have the authors made all data and (if applicable) computational code underlying the findings in their manuscript fully available?**

Reviewer #1: Yes

PLOS authors have the option to publish the peer review history of their article (what does this mean?). If published, this will include your full peer review and any attached files.

Reviewer #1: **Yes: **Jesse Geerts

Figure Files:

Data Requirements:

Reproducibility:

References:

---

## [Editor Report · Decision Letter 2]

18 Oct 2022

Dear Ms. Vijayabaskaran,

We are pleased to inform you that your manuscript 'Navigation task and action space drive the emergence of egocentric and allocentric spatial representations' has been provisionally accepted for publication in PLOS Computational Biology.

Best regards,

Daniel Bush

Academic Editor

PLOS Computational Biology

Samuel Gershman

Section Editor

PLOS Computational Biology

---

## [Editor Report · Acceptance letter]

25 Oct 2022

PCOMPBIOL-D-22-00936R2 

Navigation task and action space drive the emergence of egocentric and allocentric spatial representations

Dear Dr Vijayabaskaran,

I am pleased to inform you that your manuscript has been formally accepted for publication in PLOS Computational Biology. Your manuscript is now with our production department and you will be notified of the publication date in due course.

With kind regards,

Zsofia Freund
